# Utilizing computer vision for facial behavior analysis in schizophrenia studies: A systematic review

Zifan Jiang[1,2]*, Mark Luskus[5], Salman Seyedi[1], Emily L. Griner[3], Ali Bahrami Rad[1], Gari D. Clifford[1,2], Mina Boazak[4], Robert O. Cotes[3]

1 Department of Biomedical Informatics, Emory University School of Medicine, Atlanta, GA, United States of America, 2 Department of Biomedical Engineering, Georgia Institute of Technology and Emory University, Atlanta, GA, United States of America, 3 Department of Psychiatry and Behavioral Sciences, Emory University School of Medicine, Atlanta, GA, United States of America, 4 Animo Sano Psychiatry, PLLC, Durham, NC, United States of America, 5 Department of Psychiatry and Behavioral Sciences, University of California, San Francisco, San Francisco, CA, United States of America

* zifanjiang@gatech.edu

## Abstract

### Background

Schizophrenia is a severe psychiatric disorder that causes significant social and functional impairment. Currently, the diagnosis of schizophrenia is based on information gleaned from the patient's self-report, what the clinician observes directly, and what the clinician gathers from collateral informants, but these elements are prone to subjectivity. Utilizing computer vision to measure facial expressions is a promising approach to adding more objectivity in the evaluation and diagnosis of schizophrenia.

### Method

We conducted a systematic review using PubMed and Google Scholar. Relevant publications published before (including) December 2021 were identified and evaluated for inclusion. The objective was to conduct a systematic review of computer vision for facial behavior analysis in schizophrenia studies, the clinical findings, and the corresponding data processing and machine learning methods.

### Results

Seventeen studies published between 2007 to 2021 were included, with an increasing trend in the number of publications over time. Only 14 articles used interviews to collect data, of which different combinations of passive to evoked, unstructured to structured interviews were used. Various types of hardware were adopted and different types of visual data were collected. Commercial, open-access, and in-house developed models were used to recognize facial behaviors, where frame-level and subject-level features were extracted. Statistical tests and evaluation metrics varied across studies. The number of subjects ranged from 2-120, with an average of 38. Overall, facial behaviors appear to have a role in estimating

**Data Availability Statement:** All relevant data are within the paper and its Supporting information files.

**Funding:** Research reported in this publication was supported in part by Imagine, Innovate and Impact (I3) Funds from the Emory School of Medicine and through the Georgia CTSA NIH award (UL1-TR002378). The funders had no role in study design, data collection and analysis, decision to publish, or preparation of the manuscript.

**Competing interests:** NO authors have competing interests.

diagnosis of schizophrenia and psychotic symptoms. When studies were evaluated with a quality assessment checklist, most had a low reporting quality.

## Conclusion

Despite the rapid development of computer vision techniques, there are relatively few studies that have applied this technology to schizophrenia research. There was considerable variation in the clinical paradigm and analytic techniques used. Further research is needed to identify and develop standardized practices, which will help to promote further advances in the field.

## Introduction

Schizophrenia is a severe psychiatric disorder with a lifetime prevalence of approximately 0.48% [1]. This conditionis slightly more common in males [2], appears generally during early adulthood [3], and causes significant social and functional impairment. In 2013, schizophrenia was thought to have an annual economic burden of $155 billion in the United States [4]. Since the identification of schizophrenia in the late 1800s, significant efforts have been made to characterize symptoms of the disorder. The 5th edition of the Diagnostic and Statistical Manual of Mental Disorders (DSM-5) indicates that for a diagnosis of schizophrenia to be made, two or more of five symptom categories must be present [5]. These five symptom categories include delusions, hallucinations, disorganized speech, disorganized behavior, and negative symptoms [5]. Schizophrenia is an illness that demonstrates heterogeneity in its symptoms from person to person, and each of these symptom categories can vary vastly by presentation leading to overlap with other diagnoses. Additionally, schizophrenia has a heterogeneous longitudinal course, with some individuals having a relapsing remitting course, others chronic symptoms, and others with symptoms followed by remission [6].

Currently, the diagnosis of schizophrenia is based on the self-report of the patient, what the interviewer observes, and collateral information, all of which can be highly subjective. Reducing subjectivity in establishing the diagnosis of schizophrenia is necessary from both a research perspective (to ensure treatments work for people with the same underlying condition) and a clinical perspective. Clinically, many people with schizophrenia have a lack of awareness that they have an illness [7], and those with poor insight may be at risk for nonadherence to antipsychotic medications and other negative outcomes [8]. Clinicians often are easily able to identify and evaluate positive symptoms of the illness, but may struggle more with the identification, assessment, and quantification of negative symptoms [9]. Negative symptoms contribute to the overall disability of the illness more than positive symptoms, and include symptoms such as a lack of motivation, social withdrawal, alogia (poverty of speech), and affective flatting. Affective flatting is defined by diminished emotional expressivity in the face, is an example of a negative symptom with the potential to objectively quantify. Despite recognition of impairment of facial expressions as a key diagnostic construct in schizophrenia, research in this area has been limited, and the most recent reviews on the topic are nearly two decades old [10, 11].

Early analyses of facial expressions were primarily conducted using the Facial Action Coding System (FACS). FACS, developed by Ekman and Friesen in 1978, is a framework for developing objective and repeatable methods of coding of facial movement [12]. The system relies on trained rater coding of the presence and magnitude of multiple facial action units (AUs) such as facial, eye, and head movements. A visual illustration can be found in [13]. These

ratings formalized the methodology for the evaluation of subject facial movement and expression. In the case of schizophrenia, the system allowed for the identification of variation in patient expression in negative emotionssuch as sadness and anger [14], reduction in patient expression of happiness [10], reduction in patient emotional expressions [3, 15], and reductions in facial responsivity [16]. Still, despite its strengths, FACS was an expensive and manually laborious methodology.

Since the turn of the 21st century, machine learning has played an increasing role in mental health research [17], which is driven by rapid development of affective computing, increases in computing power, and ease of data acquisition. Incremental algorithm improvements progressing from the single layer perceptron to the convolutional neural network have also led to significant advances in the field of computer vision [18, 19]. The combination of these factors has led to the development of automated FACS software, such as Noldus FaceReader [20] and Openface 2.0 [21]. In regard to the FACS, the present state-of-the-art model [22] classifies AUs with F1-score and accuracy values of 0.55 and 0.91 respectively in the testing set of the EmotioNet dataset [23], consisting of 23 AUs presenting in 200,000 images.

Given the potential applicability of these models, researchers have used them to evaluate facial expressivity. Facial expression models have been studied in depression [24, 25], autism [26], dementia [27, 28], and schizophrenia research. Despite the increasing adoption of this technology, a review of the present state of computer vision models in understanding facial expressions in schizophrenia has not been conducted. Previous reviews have investigated the usage of computer-vision-based facial information in medical applications in general [29, 30], but they focused more on the specific technical facial analyses adopted than the complete processing and analyzing pipeline, and few schizophrenia studies were discussed in detail.

Here, we conduct a systematic review on the use of computer vision in the evaluation of facial expressivity in schizophrenia. A systematic narrative synthesis will be provided with the information presented in the text and tables to summarize and explain the characteristics and findings of the included studies. The narrative synthesis will describe the current work, its evolution, and clinical findings, in addition to discussing the data processing pipeline.

## Methods

### Searching methods

We conducted a literature searchfor publications published before (including) December 2021, on Google Scholar and PubMed in February of 2022 using the following search terms: ("facial emotion" OR "facial expression" OR "facial analysis" OR "facial behavior" OR "facial action units") AND "schizophrenia" AND "computer vision". Multiple synonyms and sub-categories of facial behaviors were used in the first keyword set to cover a broad definition of facial behaviors, and the latter two keywords were selected to limit our search with studies that used computer vision in schizophrenia. Based on the articles we found in the search, we conducted a secondary search to include othernotable and relevant papers worthy of inclusion, which were written by the same group of authors, and included relevant articles which were cited by articles found in the primary search. The secondary process was adopted to enhance the review by including those relevant articles that were not discovered in the first search using a general search process. The detailed process can be found in the PRISMA flow diagram in Fig 1.

### Inclusion and exclusion

The inclusion and exclusion process is shown in Fig 1. Returned records were first filtered by the presence of related keywords in the titles, where titles without any related keyword were

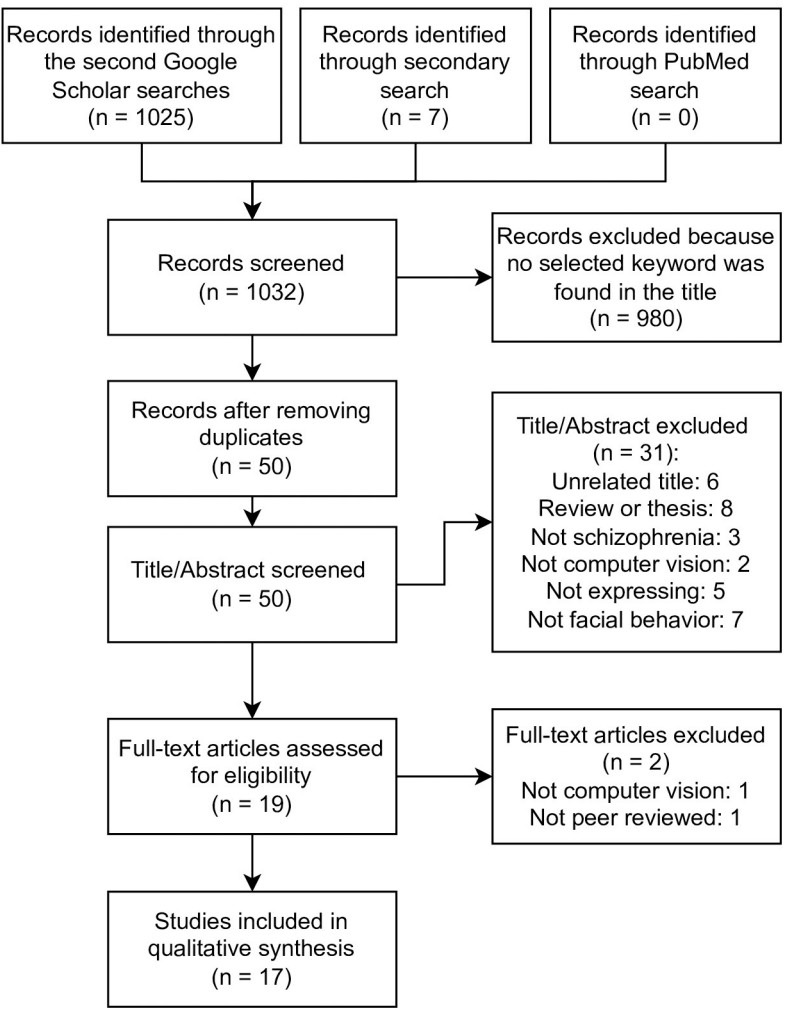

**Fig 1. PRISMA flow diagram.**

excluded. Those keywords include but not limited to "schizo-", "psychosis", "schizophrenia", "schizophrenic", "psychiatric/psychiatry" and "neuropsychiatric/neuropsychiatry". Then the duplicated records were removed. After that, 45 records were screened by titles and abstracts, where 26 records were excluded because their titles were unrelated to the surveyed topic, they did not represent original research (thesis or review), because they did not related to schizophrenia (not schizophrenia), because they only consisted of human-rated subject affect or facial movement (not computer vision), or because they focused on the processing instead of the expression of facial behaviors (not expressing). Lastly, 19 records were assessed for eligibility where one was excluded because it did not relate to schizophrenia and another one was excluded because it was not peer-reviewed and had limited rigor.

## Data collection

A data collection survey was conducted by ZJ and ML to extract relevant data, including research goals, findings, interview types, interview structures, hardware used, types of data captured/used, data pre-processing method, behavioral features, statistical testing method, and model evaluation method. For the purposes of this review, we specifically report on clinical

processes and data handling components that we discovered within the literature. From the clinical perspective, we report on how computer vision is currently used in the evaluation of schizophrenia, the general clinical findings in relation to those uses, and the interview structure during which data is collected. From the perspective of data handling, we evaluate the data pre-processing, the data processing, and hardware used in the literature.

In addition, the reporting quality of the studies were assessed with Transparent Reporting of a multivariable prediction model for Individual Prognosis Or Diagnosis (TRIPOD) [31]. The completed quality assessment can be found in S2 Table.

## Results

Seventeen articles were selected for inclusion in this review. These articles were published from 2007 to 2021, with the number of publications increasing over time: three completed before 2009 [32–35], four completed between 2010 to 2014 [36–39], and eight were completed subsequent to 2015 [40–48].

### Study objectives and participant characteristics

The study objectives were divided into three types: 1) descriptive, meaning those that described schizophrenia phenomenology, 2) predictive, meaning those that utilized predictors to classify presence or absence of schizophrenia or those which predicted certain clinical outcome scores based on facial expressions, or 3) those which included both descriptive and predictive outcomes. Of the 17 included studies, eight were descriptive, one was predictive only, and eight were descriptive and predictive. The study objective type, participant characteristics, description of the studies, and a summary of the findings can be found in Table 1. Of the descriptive studies, five used automated computer vision techniques to describe schizophrenia emotional expression and three to describe specific movements, such as facial movement, head movement hand movement or body movement. Of those that included predictive components, three were used to predict schizophrenia presence or absence, and four to predict schizophrenia severity using clinical outcome measures. The number of participants included in the studies ranged from ranged from two (case study, one control and one patient) to 120, with an average of 38. Four studies did not include a control.

### Interview techniques

Only 14 articles used interviews during the data capture (video recording) phase as shown in Table 2. For those that used an interview, the interview structure can be broadly broken into two classifications: evoked and passive. In evoked interviews, participants were asked to express certain emotions, such as anger and happiness. In passive interviews, participants were asked a series of questions, and their resulting facial expressions were recorded. Of the 14 studies, four adopted an evoked approach, seven used a passive approach, and three used a combination of interview styles. Of the 11 that used a passive interview, all were novel interviews developed by the authors. Of these, six were structured, four were semi-structured, and one was unstructured. Over the years, the articles are utilizing progressively more passive approaches.

The evoked interview style was conducted so that authors could associate different expressions with schizophrenia symptom burden. Alvino et al. [32] used an evoked interview to determine that patients with schizophrenia demonstrated particular deficiencies in expressing anger when compared to healthy controls. Wang et al. [33], using both an evoked and passive approach, identified disgust as an emotion with decreased expressivity in patients with schizophrenia.

**Table 1. Overview of the participants, objective types, descriptions and findings.**

| Article | Year | Subject | Type | Description | Findings |
|---|---|---|---|---|---|
| [32] | 2007 | 11 SZ, 10 NC | Descriptive | Developed a computational framework to quantify intended emotional expression differences between patients with schizophrenia and healthy controls matched for age, ethnicity, and gender. | Significant difference in average abilities to express emotions, especially in the case of anger. The average abilities to express emotions correlated significantly with clinical severity of flat affect. |
| [33] | 2007 | 12 SZ, 12 NC | Descriptive | Provided a framework to quantify the facial expression abnormality of patients with schizophrenia in posed and evoked emotions by combing 2D and 3D facial features and compared with results from human raters. | Human raters can only correctly identify a low percentage (mostly 40% to 70% except for happiness) of intended emotions for both controls and patients, but showed different accuracies for controls and schizophrenia patients. Significant group difference in evoked disgust was found. |
| [34] | 2007 | 12 SZ, 12 NC | Descriptive | Captured facial expressions of individuals and quantified their expression flatness by estimating overlap between different facial expression clusters in the learned embedding. | Patient group has much larger facial expression overlap than the control group, and demonstrate that the flat affect is an important symptom in diagnosing schizophrenia patients. |
| [35] | 2008 | 1 SZ, 1 NC | Descriptive | Created an automated computerized scoring system as an alternative to FACS for systematic analysis of facial expressions of healthy controls, schizophrenia patients and patients with Asperger's syndrome. | The healthy control expressed intended emotion better than the patient with Asperger's and schizophrenia (especially in the fear). The control has more neutral expression than the two patients. |
| [38] | 2010 | 27 SZ, unreported number of NC | Descriptive | Authors aimed to determine whether automated video-based quantification of body movement could be reliable indicators for nonverbal behavior in schizophrenia patients, and if body movement is valid as a measure of expressiveness. | Automated MEA-based detection of body and head movement and movement speed was found to be highly reliable, with clear indications for its validity. MEA provides an objective assessment of body movement. |
| [36] | 2011 | 4 SZ, 4 NC | Descriptive | Developed an automated FACS based on advanced computer science technology and derived quantitative measures of flat and inappropriate facial affect automatically from temporal AU profiles. | NA |
| [39] | 2013 | 20 SZ, 100NC | Descriptive and predictive | Determined whether schizophrenia patients display less speaking gestures and listener nods and whether patients' increased symptom severity and poorer social cognition are associated with patients' reduced gesture and nods. Additionally, authors aimed to determine if patients' partners compensate for patients' reduced nonverbal behavior by gesturing more when speaking and nodding when listening. | Patients with schizophrenia exhibit reduced rates of gesture making compared to healthy controls. Increased levels of negative symptoms are associated with poorer rapport with patients. |
| [37] | 2014 | 28 SZ, 26 NC | Descriptive and predictive | The authors worked to develop novel measures of facial expressivity using information theory. In particular, they developed measures of ambiguity and distinctiveness in facial expressivity, and hoped that these measures could be used to analyze large data sets of dynamic expressions. | Results indicated that ambiguity and distinctiveness of expression were both associated with a diagnosis of schizophrenia. The method developed is more repeatable and objective than observer-based rating scales. Predictions were highest for measures of overall facial expression, with an F-score of 12. |
| [45] | 2015 | 34 SZ, 33 NC | Descriptive and predictive | This study aimed to pair data-driven analysis of facial expression with descriptive methods using machine learning tools and other technology. | Results from this study are in agreement with previous studies, which demonstrate that schizophrenia symptoms result in changes to AUs when compared to healthy controls. |
| [47] | 2016 | 34 SZ, 33 NC | Descriptive and predictive | The authors aimed to create 'prototype' facial expression clusters in order to study a wider range of facial features than traditional AU and FACS computation allows for. | The authors findings were consistent with prior studies, which showed that schizophrenia patients overall have lower levels of facial expressivity. |
| [46] | 2016 | 34 SZ, 33 NC | Descriptive | The authors aimed to compute discriminative features of AU activity for the purpose of measuring the following qualities, which represent symptomology used in the diagnosis of schizophrenia: flat affect, incongruent affect, and inappropriate affect. | In contrast with previous studies, the authors found that patients with schizophrenia exhibited reduced amounts of expression in positive emotional responses. Their findings also suggest that the magnitude of changes in facial expression may correlate to symptom severity. |

*(Continued)*

**Table 1.** (Continued)

| Article | Year | Subject | Type | Description | Findings |
|---|---|---|---|---|---|
| [44] | 2016 | 18 SZ | Descriptive and predictive | Overarching goal was to create novel methods for examining clinical behavior by identifying behavioral indicators relevant to various symptoms. Application to psychiatric populations could provide needed method to collect objective behavioral data. Authors worked to identify behavior indicators relevant to certain psychosis symptoms as measured by clinical scales and determine which structured interview questions correlate to facial findings suggestive of specific psychotic symptoms. | Negative and positive symptoms are best elicited via different questions. E.g. positive symptoms were elicited via questions regarding the patient's energy, and negative symptoms were elicited via questions regarding self-confidence. AU5 and AU6 are activated more frequently in patients with depression. AU12 is negatively correlated with the PANSS Negative summative scale. Overall conclusion was that AUs can be used to detect psychotic symptoms as measured on the PANSS, BPRS, and MADRS. There is value at evaluating facial expressions at the question level. |
| [43] | 2017 | 1 SZ, 1 NC | descriptive | Compared facial expressions of a patient with schizophrenia and a healthy control, utilizing marker-based technology that recognizes facial features. | Facial expressivity intensity was higher in the healthy control and analysis of facial expressions using marker-based technology displays high fidelity. |
| [42] | 2018 | 91 SZ | Descriptive and predictive | Proposed SchiNet, a novel neural network architecture, trained on large-sclae FACS datasets, that estimates presence and intensity of action units. Then it is used to predict expression-related symptoms from two commonly-used assessment interviews; Positive and Negative Syndrome Scale (PANSS) and Clinical Assessment Interview for Negative Symptoms (CAINS). | Significant correlations are found between symptoms and the frequency of occurrence of automatically detected facial expression. The score of several symptoms in the PANSS and CAINS interviews can be estimated with a MAE less than 1 level. Automatic estimation of symptom severity needs further improvement to reach human level performance. |
| [40] | 2019 | 25 SZ | Descriptive and predictive | Develop a proof-of-concept for the potential of using the machine learning FAR system as a clinician-supporting tool, in an attempt to improve the consistency and reliability of mental status examination. | There is a lack of inter-rater reliability between five senior adult psychiatrists working in the same mental health center. Automatic facial analysis may be able to predict the label provided by psychiatrists. |
| [41] | 2019 | 74 SZ | Predictive | Incorporated temporal information into the SchiNet using stacked GRU to directly addresses the problem of Treatment Outcome Estimation (TOE) in schizophrenia—more specifically, is aimed at determining whether specific symptoms have improved or not by analysing jointly two videos of the same patient, one before and one after the treatment. | Proposed method can determine The TOE of CAINS expression symptoms and PANSS negative symptoms with an accuracy of about 0.7 (0.64–0.71) and a F1 score of around 0.4 (0.33–0.46). The determination is more accurate with proposed specifically designed TOE method than applying symptom severity estimation to before and after treatment independently. |
| [48] | 2021 | 18 SZ, 9 NC | Descriptive and predictive | Developed a remote, smartphone based assessments to capture objective measurement of head movement, which were then used as features to predict both PANSS subscale scores and individual items in each of those subscales, with age and gender as confounding variables. | Head movements acquired remotely through smartphone were able to classify schizophrenia diagnosis and quantify symptom severity in patients with schizophrenia. |

SZ = schizophrenia patient, NC = normal control.

In Tron et al. [45, 47], participants were asked, "Tell me about yourself", which was followed with three questions that intended to lead to the expression of emotions. Vijay et al. [44] interviewed patients in a semi-structured naturalistic clinical encounter that used 13 standardized questions, such as, "How has your mood been?" and "How is your energy?". In addition to the diversity of the interview techniques, detailed descriptions of the interviews were unavailable. Several papers stated that they asked specific questions but did not report the details of the questions in the manuscript.

## Hardware

Hardware for data collection varied dramatically over the period of the review, often reflecting changes in the quality and pricing of consumer technology. Due to the low quality of consumer-level systems, early studies used multi-camera systems [33, 45–47] and physical markers placed on the subjects face for landmark localization [39, 43]. In contrast, more recent studies

**Table 2. Overview of participant interviews.**

| Article | Passive/ Evoked | Interview Structure |
|---|---|---|
| [32] | Evoked | Subjects were asked to make facial expressions of happiness, sadness, anger, and fear. |
| [33–35] | Both | Participants were asked to express happiness, anger, fear, sadness, and disgust at mild, moderate, and peak levels, respectively. In the evoked session, participants were guided through vignettes, which were provided by the participants themselves and describe a situation in their life pertaining to each emotion. |
| [38] | Passive | Researchers conducted role-play tests (RPT), which were used to measure social competence in schizophrenia. All RPTs were video recorded. Each test consisted of 14 social scenes that represented three response domains. |
| [36] | Evoked | Researchers had participants express the following emotions: happiness, sadness, anger, fear, and disgust. |
| [37] | Both | Patients were recorded expressing sadness, anger, happiness, fear, and disgust. Each emotion recording lasted for approximately 2 minutes. Additionally, patients were recorded while being read self-recorded vignettes about times in their life in which they experienced these emotions. |
| [45, 47] | Passive | The interview was semi-structured and involved a single question of "Tell me about yourself" followed by three emotionally evocative questions that were not described. |
| [46] | Unknown | Participants underwent a short, structured interview that was not described by the authors. |
| [44] | Passive | Participants were interviewed in a style consistent with a routine clinical encounter for a patient under inpatient treatment for schizophrenia. The interview was semi-structured, consisted of 13 questions, and was approximately 10 minutes in length. |
| [41, 42] | Passive | Recordings from a previous [49] trial were used. No novel participant interviews in this study. |
| [40] | Passive | Participants underwent a semi-structured 10-minute interview that consisted of the following ten questions: (1) Can you please present yourself and tell me a bit about yourself? (2) How do you feel? (3) Can you tell me about the events that led to your current hospitalization? (4) Can you tell me some things about your family? (5) Can you tell me of something sad that has recently happened to you? (6) Can you tell me of something pleasing that has recently happened to you? (7) Is there anything else you want to add? (8) What do you think about the recent situation in the country? (9) What are your future plans? (10) How did you feel about talking with me in front of the camera? |
| [48] | Evoked | Open-ended questions such as "What have you been doing for the past few hours?" and "What are your plans for the rest of the day?" were asked to elicit a free verbal response. |

used one visible-light camera, an approach which may be more feasible in a clinical setting, and more cost-effective. Later works also had the benefit of the discoveries of early works using higher quality systems. Among those who reported the video or image collection details, the resolution ranged from 1280x960 pixels [44] to 1920x1080 pixels [42] and the video collection rate ranged from 25 [42] to 30 [44] frames-per-second. Of those that used 3-dimensional data, one group [33] used polyocular stereo cameras and a color camera, while another [45–47] used 3D cameras based on structured light technology.

## Data pipeline

Table 3 summarizes how data was collected, processed, analyzed and reported in the 17 studies we surveyed. More specifically, the following aspects were included: (1) the types of data collected, (2) features calculated from the data, (3) and the corresponding statistical analyses and performance metrics used for reporting the results. A visualization of the adopted data pipelines can be found in Fig 2. Detailed data processing steps can be found in S1 Table.

**Types of the visual data.** Raw data inputs of all included studies were vision-based. Table 3 shows the type(s) of data used in each study. Five types of visual data were used,

**Table 3. Overview of data processing and statistical analyses.**

| Article | Frame-level Features | Subject-level Features | Statistical Tests | Performance Metrics | Validation |
|---|---|---|---|---|---|
| | | Studies with 2D or 3D image data | | | |
| [32] | SVM output of the intended expression normalized with outputs from other SVMs. | Average normalized output. | Paired t-test | PCC | NA |
| [33] | 2D features: the area of facial regions, the distance between some fiducial points; 3D Curvature Features and 3D Gabor moment invariants for six facial regions. | Lower dimensional embedding of the frame level features was learned with the ISOMAP manifold learning algorithm. | Paired t-test | NA | NA |
| | | Studies with video data | | | |
| [34] | Geometric features similar to [32]. | Lower dimensional embedding of the frame level features was learned with the ISOMAP manifold learning algorithm. A "Flatness Index" was defined as the minimal pair-wise overlap between one expression to other expressions in the ISOMAP embedding. | Paired t-test | NA | NA |
| [38] | Motion energy: the amount of grayscale changes from one frame to the next in the ROIs normalized by ROI size. | Percentage of time with detectable movement in ROIs and the speed of body movement. | paired t-tests; ANOVA | PCC; Cronbach's alpha | NA |
| [36] | Confidence and presence of the 15 AUs. | Frequency (percentage of frames presented) of single AUs and AU combination; Flatness measure: frequency of neutral frames (no AU was present); Inappropriateness measure: frequency of "disqualifying" AUs defined in [15]. | NA (method paper) | NA | NA |
| [37] | Confidence and presence of the 15 AUs. | Same as [36]. | two-way ANOVA; | PCC; Cohen's d | None |
| [44] | Intensities and presence of the 20 AUs. | Mean and standard deviations of intensities of each AU during answers to specific questions. | NA | PCC | LOSO |
| [42] | Normalized intensities of ten AUs and smile. | The Fisher vector representation of the distribution of intensities over time, from unsupervised learned Gaussian Mixture Model. | NA | SCC, PCC, MAE, RMSE | LOSO |
| [40] | Intensities of seven emotions: norm, anger, disgust, fear, happiness, sadness and surprise; Mean grey scale of the face. | Mean intensity of the emotions; Number of transitions of emotions; Standard deviation of mean gray scale. | NA | ACC | LOSO |
| [41] | Normalized intensities of ten AUs and smile. | Two stacked GRUs were used to extract clip-level (15s segments of the videos) and patient-level representations. | NA | F1, ACC | LOSO |
| [48] | Head location of each subject relative to the camera. | Average head movement. | t-test | $R^2$, Adj. $R^2$ | None |
| | | Studies with IR videos | | | |
| [39] | Identities of listener and speaker; Head and hand locations of each subject. | Head and hand movement rate; Percentage of time spent in speaking, nodding/gesture as listener of the patients, patients' partners and controls. | t-test | QICC, SE | None |
| [43] | 3D locations of the facial markers. | Average value of distances traveled by markers during shifts from a neutral position. | NA | NA | NA |
| | | Studies with depth camera videos | | | |
| [35] | Output of the five SVMs trained for classifying five expressions (happiness, sadness, anger, fear and neutral). | Output of SVMs were modeled as the observed variable in HMM, where the hidden variable indicates emotions. Four features were used: 1. the average of posterior probabilities of intended and neutral emotions; 2. the occurrence frequency of the appropriate and neutral expressions. | NA (method paper) | NA | NA |
| [45] | Activity level of each AU. | Activation Ratio: Fraction of segment during which the AU was activated; Activation Level: Mean intensity of AU activation; Activation Length: Number of frames that the AU activation lasted; Change Ratio: fraction of the period of AU activation when there was a change in activity level; Fast Change Ratio: fraction of fast changes in activation level. | One-way ANOVA, t-test | AUC, PCC | LOSO |

(*Continued*)

**Table 3.** (Continued)

| Article | Frame-level Features | Subject-level Features | Statistical Tests | Performance Metrics | Validation |
|---|---|---|---|---|---|
| [47] | Activity level of each AU. | Richness: how many prototype expressions appeared; Typicality: how similar they were to the prototype. Distribution: which expressions were more prevalent. | Bonferroni correction, one-way ANOVA, t-test | PCC, AUC | LOSO |
| [46] | Activity level of each AU. | Flatness Measures: the sum of the variance in facial activity for similarly/differently rated photos; Congruity Measures: the ratio between the sum of the variance within similarly rated photos and the sum of total variance; Inappropriateness measure: the sum of the squared difference between the average facial activity of all controls and each subject's individual facial activities. | t-test | PCC, Cohen's d | LOSO |

ACC = accuracy, Adj. $R^2$ = adjusted R square, ANOVA = analysis of variance, AU = action unit, GRU = gated recurrent units, HMM = hidden Markov model, ISOMAP = isometric mapping, LOSO = leave-one-(subject)-out, MAE = mean absolute error, PCC = Pearson correlation coefficient, QICC = corrected quasi-likelihood under independence model criterion, RMSE = root mean square error, ROI = region of interest, SCC = Spearman rank-order correlation coefficient, SE = standard error. "NA" in the Statistical Tests column indicates that no statistical test was used or was clearly reported. "NA" in the validation column indicates that no classification or regression was conducted in the study, hence the validation was not needed.

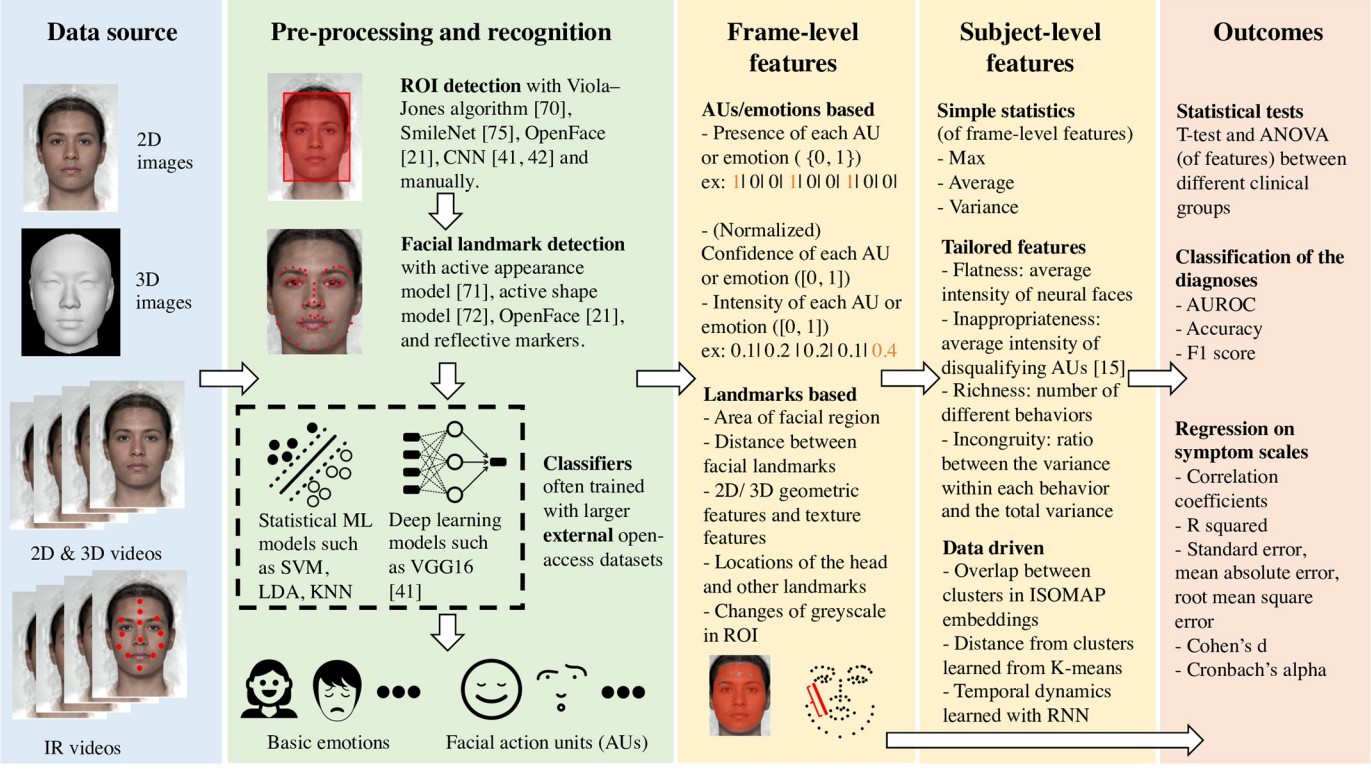

**Fig 2. Visualization of the data pipelines.** Different combinations of the methods in each section were adopted in different studies. Pre-processing and recognition methods used in commercial software were not included due to the lack of clarify on what algorithms were used in them. The face used in illustration was an average face generated from http://faceresearch.org/demos/average, which is available open access (CC-BY-4.0) [50]. The icons used in the figure are available open access (CC-BY) from the NounProject.com. 2D/3D: two/three-dimension, ANOVA: analysis of variance, AUROC: area under the receiver operating characteristic, CNN: convolutional neural network, IR: infrared, ISOMAP: isometric mapping, KNN: k nearest neighbor, LDA: linear discriminant analysis, ML: machine learning, RNN: recurrent neural network, ROI: region of interest, SVM: support vector machine.

namely two-dimensional (2D) images, three-dimensional (3D) images, 2D videos (with infrared or visible light), and 3D surface videos from structured light cameras. The earliest two studies [32, 33] used only image data, where [33] used both 2D and 3D images, and [32] used 2D images. Four studies [35, 45–47] used 3D videos, two studies [39, 43] used 2D infrared light videos, and the remaining nine studies used 2D visible light videos. Only one group [42] evaluated the advantages of using videos over images, demonstrating superior performance in both AU recognition and symptom estimation with video data.

**Behavior recognition methods.** While most studies focused on analyzing facial expressions, three [38, 39, 48] selected head movement as the primary behavior to be investigated. Kupper et al. [38] used the head as the region of interest (ROI) and used the changes of frame-wise pixel intensity in the ROI as a surrogate for head movement, while Lavelle et al. [39] adopted the vertical distance between the positions of the reflective marker on the head in consecutive frames as the movement of the head. Though both analyses were mostly based on computer vision, some level of manual help was involved, where the former requires manual selection of the head area as ROI, and the latter requires the researchers to put reflective markers on the participants. Abbas et al. [48] addressed this issue by recognizing the head area with a convolutional neural network (CNN).

The remaining 14 studies were divided into three categories based on how facial expressions were evaluated: 1) analysis of basic emotions, 2) AU analysis, and 3) surrogate measures to measure facial expression. Three studies analyzed basic emotions [32, 35, 40], including emotional categories such as happiness [32, 35, 40], sadness [32, 35, 40], anger [32, 35, 40], surprise [40], disgust [40], fear [32, 35, 40], and neutrality [35, 40]. Eight studies [36, 37, 41, 42, 44–47] focused on AU analysis, a proxy measure for underlying facial muscle movement, based on the earlier mentioned FACS. AUs including AU0 (Neutral Face [41, 42]), AU1 (Inner Brow Raiser [36, 37, 41, 42, 44–47]), AU2 (Outer Brow Raiser [36, 37, 41, 42, 44–47]), AU4 (Brow Lowerer [36, 37, 41, 42, 44]), AU5 (Upper Lid Raiser [36, 37, 41, 42, 44]), AU6 (Cheek Raiser [36, 37, 41, 42, 44]), AU7 (Lid Tightener [36, 37, 44]), AU9 (Nose Wrinkler [36, 37, 44]), AU10 (Upper Lip Raiser [36, 37, 44]), AU12 (Lip Corner Puller [36, 37, 41, 42, 44–47]), AU 14(Dimpler [44–47]), AU15 (Lip Corner Depressor [36, 37, 44–47]), AU17 (Chin Raiser [36, 37, 44–47]), AU18 (Lip Puckerer [36]), AU20 (Lip stretcher [36, 37, 44–47]), AU23 (Lip Tightener [36, 37, 41, 42, 44]), AU25 (Lips part [36, 37, 41, 42, 44–47]), AU26 (Jaw Drop [44–47]), AU 43 (Eyes Closed [41, 42, 45–47]), AU44 (Squint, [45–47]) AU45 (Blink [44]), and AU62 (Eyes Turns Right [45–47]). Other non-traditional AUs, such as smile, frown and sneer, were also mentioned in [45–47]. The remaining three studies [33, 34, 43] used surrogate measures to represent the facial expression: two studies [33, 34] only used basic emotion classification as a task to evaluate the effectiveness of the calculated features. These features, instead of classified emotions, were directly used as the quantification of participants; Another study [43] defined the average value of distances traveled by facial markers during shifts from a neutral position as the facial expression intensity.

To recognize facial expressions, six studies used existing solutions to estimate the presence and the intensity of the AUs or to locate the faces, including commercial software (FaceShift [45–47], Vicon Blade software [43]) and open-source software [44, 48] (Openface [21]). Nine studies used novel facial expression recognition (FER) methods. For the six studies before 2017, face and facial landmarks were first collected. Detected landmark locations were then used for AU recognition or emotional recognition. The combination of geometric and wavelet-based texture features and various statistical supervised classifiers (including KNN, SVM, Adaboost) had been most adopted for these AU and emotional detectors until the introduction of a CNN based end-to-end AU classifier in [42]. It is worth noting that only [35] incorporated

temporal information into facial expression recognition, and they modeled the dependencies between frames with a Hidden Markov Model.

**Analysis of frame-level and subject-level features.** The "Frame/Subject-level Features" columns in Table 3 shows the different features extracted at both frame-level and subject-level in each study. Features extracted at frame-level helped summarize the information generated from the behavior recognition in each frame or image, and features extracted at subject-level were used to represent the characteristics of a subject by summarizing and reducing the dimension of the frame-level features. This was useful for identifying meaningful patterns from a limited number of subjects. They were either used to quantitatively describe the behaviors of the subjects or used as the input for the final task, such as diagnosis classification or symptom score regression.

Although the subject-level features used in the studies focusing on facial expressions were diverse, the frame-level features appeared to primarily measure the intensity and/or the presence of a specific facial expression. Only Wang et al. [33–35] adopted the 2D and 3D geometric and texture features directly without using them to recognize the facial expression in the pre-processing steps. Additionally, AUs were only identified at the level of video clips instead of frame-level in two studies because of the protocol used by human FACS raters.

Many simple statistics of the time series of the frame-level features were used as subject-level features, including max [51], average [32, 35, 37–40, 44, 45] and variance [44, 46] of either presence or intensity of the frame-level features, often normalized when counting presence. Those statistics were also calculated in earlier studies where facial expressions were manually recognized. For example, average was used in [52, 53]. When statistics were only calculated for a specific subset of frame-level features, like neutral or disqualifying AU [53] they could be interpreted as surrogate measures of flatness and inappropriateness, respectively.

Other tailored subject-level features have also been reported. Some counted the number of different AUs or emotions expressed and defined it as richness [37, 47, 53]. Another type of feature aimed to measure how much the facial expressions alternated by calculating the percentage of time when there are changes in intensities [45]. The measure of incongruity was first introduced in [46] and defined as the ratio between the variance within each emotion and the total variance, indicating how consistent the facial expressions were when the similar emotional response was evoked.

In addition to manually designing subject-level features, many data-driven feature generation methods have been proposed over the years. Some treated frame-level features from all the frames as independent observations and reduced the dimension with Isomap [33] or Gaussian mixture model [42]. Wang et al. [34] then defined the "flatness" of each video as the minimal overlap between one expression cluster to clusters of other expressions in the learned Isomap embedding. Similarly, Tron et al. [47] first conducted clustering via K-means and then used the centroids of the clusters as prototype expressions to define measures like richness and typicality. Bishay et al. [41] took the research a step further and first made use of the temporal dynamics to learn subject-level representations with stacked Gated Recurrent Units (GRUs) [54].

**Evaluation methods.** The "Statistical Tests", "Performance Metrics", "Validation" and "Subjects" columns in Table 3 shows the different evaluation methods and population adopted in each study.

Statistical tests including t-test, analysis of variance were mainly adopted to evaluate the differences between different clinical groups. However, some studies [35, 36, 40–44] did not report using any statistical tests on the features or the performances.

To evaluate the performance of the proposed classification or regression approaches, Pearson correlation coefficient (PCC) or Spearman rank-order correlation coefficient (SCC) were

calculated in all studies that tried to estimate the symptom rating scales except two [39, 48], where corrected quasi-likelihood under independence model criterion (QICC) and standard error (SE) were reported in [39], and (adjusted) R square was reported in [48]. Other metrics used include Cohen's d, Cronbach's alpha, mean absolute error (MAE), and root mean square error (RMSE). For studies that targeted diagnosis classification, no unified metric was used in them. Instead, subsets of area under receiver operating characteristic (AUC), accuracy, F1 score were selected in different studies.

Although the two earliest predictive studies [37, 39] did not report their model performance on a held-out test set, all later predictive studies reported performance using the leave-one-subject-out (LOSO) cross-validation procedure.

## Study findings

Thirteen studies used computer vision to detect the presence or absence of schizophrenia and four used computer vision to predict disease severity. Of those predicting the presence of disease, authors aimed to identify decreases in global facial expressivity, as well as differences in emotions associated with schizophrenia. In regards to global reductions in facial expressivity, performance varied by measure and the sub-component of the measure.

Research groups attempting to predict disease severity identified components of facial expressivity associated with symptom severity scales. AU12 (which corresponds to the zygomatic major) is negatively correlated with the PANSS-NEG scale, with a correlation coefficient of -0.578 [44]. The overall magnitude of changes in facial expression is associated with 'Blunted Affect' on the PANSS scale, with an R-value of -0.598 [47]. In addition, one group identified 3 out of 7 PANSS-NEG symptoms, flat affect, poor rapport, and lack of spontaneity, as being associated with changes in facial expression [42]. Computer vision performed poorly on the P1-P7 items on the PANSS-POS scale, with no groups identifying statistically significant correlations between facial expression and positive symptoms. Regarding the BPRS symptom scale, one group noted that AU2 (which corresponds to the frontalis and pars lateralis muscles) is correlated with unusual thought content, with a correlation coefficient of 0.752 [44].

## Discussion

Although facial expressions can be identified with the help of trained experts [51, 53], manual identification fails to scale due to time and financial constraints and are not feasible in a busy outpatient clinic. Furthermore, due to the lack of easily reproducible standards for facial expressions, the field is yet to develop an objective consensus definition on what precisely constitutes affective flattening or other facial abnormalities in schizophrenia. Automated computer vision techniques may help to solve some of these challenges, as advances in affective computing have made it easier and cheaper to analyze a large amount of data while providing a consistent way to quantify facial behaviors. With improving technology harnessing advances in temporal and spatial granularity, computer vision based analysis has the potential to allow researchers to better understand the phenomenology of schizophrenia and differentiate those with schizophrenia from without it, and to help to subtype schizophrenia based on digital phenotypes. Additionally computer vision can objectively introduce non-verbal facial behavior data into the clinical area, allowing for clinicians to better identify negative symptoms and monitor treatment response from medication and psychosocial treatments. The systematic review serves as a road map for researchers to understand the current approaches, technical parameters, and existing challenges when using computer vision to analyze facial movements in patients with schizophrenia.

## Interview techniques

The papers reviewed used a broad array of interview techniques during video or image capture, which are broadly classified as either evoked or passive. While both techniques resulted in demonstrably significant differences in cross-group expressivity, passive techniques have been the primary modality used in the majority of studies and have been more frequently adopted in recent studies. The overarching goal in a passive interview is to capture a wide range of facial expressions, similar to what would be elicited in a clinical encounter. This interview style may provide data that is more relevant to a psychiatric appointment, helping researchers develop predictive models that are applicable to the clinical environment. Additionally, studies using a passive interview technique were able to appreciate a broader range of expressions and AUs in their participants, which may allow for more data collection.

Nonetheless, there is significant variability in the types of passive interview techniques. All groups, with the exception of Bishay et al. [41, 42], used novel group-developed semi-structured interviews during their data collection, which invariably impacts subject expressivity. It should therefore be of no surprise to learn that group descriptives of schizophrenia facial expressions and AU's differed. Even in looking to predictive models, there were within-group variations in model performance across interview sub-components (e.g., Periods of silence vs. each of the semi-structured interview questions [44]). Due to the inconsistencies in the interviews, we can not clearly state the best approach for future research, however, we can point to elements that stood out. A period of silence during the interview yielded facial expressions that correlated to positive symptoms as measured by PANSS, as well as questions regarding the patient's energy. Negative symptoms were elicited via questions regarding a patient's self confidence [44]. Positive symptoms were elicited via questions regarding the patient's energy. AU5 and AU6 were activated more frequently in patients with depression. AU12 was negatively correlated with the PANSS Negative subscale. Overall, AUs appear to have a role in estimating psychotic symptoms as measured on the PANSS, BPRS, and MADRS.

## Effect of hardware and data source

The type of hardware and collected data also varied across studies. As Table 3 shows, some studies involved expensive procedures that entailed the utilization of 3D reflective surface markers in still images. Most of the more recent approaches used less complicated equipment, with many simply utilizing visual spectrum cameras, and some utilizing infrared and depth-recording devices. Results indicated that studies with inexpensive and accessible hardware were also able to model facial behaviors successfully and demonstrated cross-group differences. While most recent studies in this area have defaulted to use inexpensive and accessible hardware, it is unclear whether or how much the performance benefits from additional information from more complicated and expensive setup. For example, the effectiveness of using 3D surface data or reflective-marker-based methods still await comparison with 2D based methods in a larger dataset. With the increasing use of telemedicine in psychiatry in recent years, even further accelerated by the COVID-19 pandemic [55], another unanswered question is whether data collected remotely (such as in [48]) can provide a similar level of information as the data collected in a lab-controlled environment.

In addition, most studies did not fully utilize the data acquired. For instance, most studies acquired video data, which enabled the generation of subject-level features like flatness and change ratio based on the facial behaviour fluctuations. However, facial behavior recognition in most studies (except [33]) was still conducted at the static image in each frame of the videos, which makes it unknown whether the use of dynamic information (using multiple frames) in videos improves the accuracy of within or across subject facial behavior recognition.

## Existing barriers and future directions

Because of the sensitive and potentially identifiable nature of facial data for patients with schizophrenia, none of the datasets mentioned in this survey are publicly available. In addition, different datasets used in different studies vary in many aspects such as data type, subject size, demographic, diagnoses distribution, and the selection of performance metrics. Consequently, it is difficult to compare the performances of the different methodologies evaluated, which adds an additional burden to the researchers who want to follow or replicate the previous studies.

Considering the population whom the analyses were applied to, this performance comparability issue is two-fold: how the methods perform in general and how they perform in the schizophrenia population, who might have significantly different facial behaviors from the population whom the algorithms were trained on. The former issue can be solved by comparing the proposed method with other recently proposed methods in the same dataset, either on the private dataset or on the publicly available dataset. Taking AU recognition as an example, dataset like EmotioNet [23] could be used as the benchmarking dataset. In addition, many state-of-the-art methods often provide publicly available implementations, such as JAA-Net (Joint facial action unit detection and face alignment via adaptive attention) [56]. The second issue, however, is much less straightforward. Most of the studies did not evaluate the performance of their methods in the patients, which could lead to an overly optimistic estimation. Results in [32] indicated that algorithms trained on healthy subjects might have significant performance differences when applying to patients and healthy controls. It is also important to explicitly note that the mismatch of distribution in demographic and cultural distribution between training and application population could lead to poor performance and bias in targeted population [57]. Besides, whether it is plausible to accurately recognize some types of the patients' facial behaviors, such as emotion, is still debatable since their behaviors might be conflicted with their intention.

An additional point for consideration include the limitations of the psychometric test raters. There can be significant inter-rater variability for psychometric tests of schizophrenia. For instance, one study reported that individual items of the PANSS had inter-rater reliabilities ranging from 0.23 to 0.88 in intraclass correlation [58]. Given that the performance of predictive models depend on the "gold standard" scored by human raters, work would need to be done to improve the accuracy and reliability of ratings. One potential solution to this challenge would include utilizing average psychometric scores from multiple raters. Besides inconsistency in psychometric tests, disparities in diagnosis are introduced by factors such as racial bias. African American and Latino American patients are diagnosed with psychotic disorders at approximately three times the rate of Euro-Americans [59]. It is imperative that inappropriate diagnoses due to racial bias are not perpetuated in the developed algorithms. Again, a potential solution in this case would be to ensure that selected cases for model training are diagnosed and rated by multiple raters.

Many studies have tried to avoid directly interpreting facial behaviors, but to use the recognized pattern as features for data-driven description or classification of the patient population. Nevertheless, it might alleviate the interpretability of the method and prevent (or delay) implementation in a clinical setting. In addition, learned subject-level features might not necessarily be utterly superior to manually designed ones. Bishay et al. [41] compared the performance of the manually designed facial behavior features from [45, 47] with the data-driven ones and showed the manual ones could be better in some cases.

As described above in the results section, temporal dynamics of the facial behaviors were not effectively used neither in behavior recognition modeling nor in the final symptom/

treatment output classification or estimation. The former might be easier to start with since there are temporal facial expression datasets publicly available, such as Annotated Facial-Expression Databases (AFEW) [60]. Recent progress in computer vision could help bring superior performance in facial expression recognition. Replacing the current computer vision models used in affective computing with better backbone neural networks like ConvNext [61] and new video classification frameworks like video vision transformer [62] could be a potential direction.

The latter issue of using temporal dynamics in schizophrenia, like other studies in psychiatry, is limited by the number of participants recruited, hence, is limited by the size of the final dataset. Therefore, the complexity of the model must be kept in mind when designing the method; otherwise, it will inevitably overfit to the training data. Bishay et al. [41] took this into account and selected GRUs [54] over a long short-term memory network (LSTM) [63] for having fewer parameters.

Another barrier in applying computer vision to schizophrenia research is the lack of the open-source, state-of-the-art computer vision toolbox that is specifically designed for psychiatric facial behavior analysis. The most widely used currently is Openface 2.0 [21] that was released in 2016. Although it covers a wide range of analysis, such as head tracking, facial AU recognition, and gaze tracking, the methods used perform significantly poorer than the latest deep learning-based methods (such as JAA-Net [56]). Furthermore, since Openface is not specifically designed for psychiatry studies, it only focused on the frame-level behavior recognition without implementing any video-level analysis. Lastly, the interface can be difficult for researchers without previous experience in programming. Therefore, the next generation of the open-source toolbox that aims to tackle these issues might help accelerate the use of computer vision in schizophrenia.

Last but not the least, how and what kind of data should be saved have not been sufficiently discussed. Ideally, just enough information should be saved for the necessary evaluation of disorders and protect the privacy of the subjects to as great an extent possible. Taking video data as an example, one type of approach is to de-identify the person in the video by pixelating or blurring certain parts of the face or trying to keep some of the behavior information while switching other facial properties via methods like deepfake [64]. The second type of approach is not to save the videos, but to maintain the derived features that can help to preserve privacy. The second approach may be more more efficient than the first, and can conserve the most relevant information without necessarily leading to a PHI leak, as shown before [65].

## Current use and the future

Present use of computer vision techniques in schizophrenia is focused on subject descriptives, prediction of disease presence, and prediction of disease severity. From a descriptive standpoint, as described in the "study findings" section of the results, these techniques have demonstrated significant differences between patient and control with respect to facial expressions. Predictive content remains to perform modestly and much work still needs to be done in this area for computer vision tools to effectively augment clinical care. Notably, modest performance of predictive models for disease severity states is not surprising. Schizophrenia impacts more than just facial expression and eye movement in patients. Much of the disorder is evaluated through a person's speech (which expresses the content of one's thoughts, and is an expression of one's thought process). It is in a person's speech that patient key characteristics are evident, including their expression of hallucinations, delusions, and disorganization in thought. Nevertheless, we were surprised to see that even for the best performing model, prediction of affective blunting (characterized by facial expression) was no better than (R

Pearson = 0.686, p ≪ 0.01) [45]. This may very well be due to the inconsistency in affective blunting predictions across raters. One group, for instance, found affective blunting to have amongst the lowest percentage of rater agreement within the PANSS [66]. With such low inter-rater agreement it would be unrealistic to expect models trained on these human-rated tools to perform much better than humans. Rather, it may be that the future of the use of computer vision for the evaluation of facial expression in schizophrenia lies in the development of a new digital biomarker or improved definitions of terms used in the psychiatric mental status (i.e. affective quality and quantity). As they stand, facial landmark predictors have achieved unparalleled performance. Inter-ocular distance estimates using these tools at the state of the art can be estimated with a normalized mean square error (NME) of 3.13 [67]. That means that estimates of the distance between outer eyebrow lids can generally be estimated with a rough 3% error margin. One group has reported human level estimates to sit at an NME of 5.6 [68]. With improving landmark prediction performance, the field is primed for exploration of the role of time series characterizations of landmark displacement as a digital biomarker. Certainly, AU predictions may also be used, but performance here has not achieved the same level as landmark prediction. Much needs to be done in order to achieve a state where landmark displacement could be utilized as a biomarker. That includes development of standard performance requirements for landmark predictors, characterization of population level norms and variations, and, in the case of schizophrenia, evaluation for differences in these biomarkers from population norms. While unlikely to independently assist in schizophrenia diagnostics, this digital biomarker may in combination with others be utilized to track disease state and treatment response. Such biomarkers may eventually be used in a manner similar to the complete blood count, or may be nested in clinical decision support tools.

## Conclusion

Here we reviewed the utilization of computer vision and affective computing techniques in schizophrenia research to date. We reported on the various uses of these techniques and those elements we felt to be relevant to researchers interested in utilizing these techniques in their work. We found that despite the rapid pace in automated facial and landmark detection techniques, there remains to be limited utilization of these techniques in the study of schizophrenia. More studies and testing on larger and more diverse population need be conducted, and standardized metrics need to be reported to enable the community to select and further develop the paradigm and methods suitable for this field. Lastly, as the progress in this field will depend on the uptake of the work by multiple research and clinical groups, we hope that this review promotes entry and work in this area.

## Supporting information

**S1 Table.**
(PDF)

**S2 Table. Assessment of study quality.**
(XLSX)

**S3 Table. TRIPOD checklist for reporting quality assessment.**
(DOCX)

**S1 Checklist. Data processing steps [21, 23, 32–48, 69–79].**
(PDF)

## Acknowledgments

We thank Scott Haden Kollins and Matthew Engelhard (both from Duke University) for designing the data collection survey utilized in this review with Mina Boazak.

## Author Contributions

**Conceptualization:** Zifan Jiang, Gari D. Clifford, Mina Boazak, Robert O. Cotes.

**Data curation:** Zifan Jiang, Mark Luskus, Mina Boazak.

**Formal analysis:** Zifan Jiang.

**Funding acquisition:** Gari D. Clifford, Mina Boazak, Robert O. Cotes.

**Methodology:** Zifan Jiang, Mark Luskus, Mina Boazak.

**Project administration:** Gari D. Clifford, Robert O. Cotes.

**Supervision:** Gari D. Clifford, Mina Boazak, Robert O. Cotes.

**Writing – original draft:** Zifan Jiang, Mark Luskus, Mina Boazak, Robert O. Cotes.

**Writing – review & editing:** Zifan Jiang, Salman Seyedi, Emily L. Griner, Ali Bahrami Rad, Gari D. Clifford, Mina Boazak, Robert O. Cotes.

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
