## [Decision Letter · Decision Letter 0]

25 Jan 2022

PONE-D-21-39628Utilizing computer vision for facial behavior analysis in schizophrenia studies: A systematic reviewPLOS ONE

Dear Dr. Jiang,

Thank you for submitting your manuscript to PLOS ONE. After careful consideration, we feel that it has merit but does not fully meet PLOS ONE’s publication criteria as it currently stands. Therefore, we invite you to submit a revised version of the manuscript that addresses the points raised during the review process.

We look forward to receiving your revised manuscript.

Kind regards,

Felix Albu, Ph.D.

Academic Editor

PLOS ONE

Journal Requirements:

2. Please attach an assessment of study quality as a Supplemental file.

"Research reported in this publication was supported in part by Imagine, Innovate and Impact (I3) Funds from the Emory School of Medicine and through the Georgia CTSA NIH award (UL1-TR002378)."

"Research reported in this publication was supported in part by Imagine, Innovate and Impact (I3) Funds from the Emory School of Medicine and through the Georgia CTSA NIH award (UL1-TR002378). We thank Scott Haden Kollins and Matthew Engelhard (both from Duke University) for designing the data collection survey utilized in this review with Mina Boazak."

We note that you have provided funding information. However, funding information should not appear in the Acknowledgments section or other areas of your manuscript. We will only publish funding information present in the Funding Statement section of the online submission form. 

"Research reported in this publication was supported in part by Imagine, Innovate and Impact (I3) Funds from the Emory School of Medicine and through the Georgia CTSA NIH award (UL1-TR002378)."

Additional Editor Comments:

The decision is Major Revision. Please address all the comments of the reviewers.

Reviewers' comments:

Reviewer's Responses to Questions

**Comments to the Author**

1. Is the manuscript technically sound, and do the data support the conclusions?

Reviewer #1: Partly

Reviewer #2: Partly

Reviewer #3: Partly

2. Has the statistical analysis been performed appropriately and rigorously? 

Reviewer #1: No

Reviewer #2: N/A

Reviewer #3: Yes

3. Have the authors made all data underlying the findings in their manuscript fully available?

Reviewer #1: Yes

Reviewer #2: Yes

Reviewer #3: Yes

4. Is the manuscript presented in an intelligible fashion and written in standard English?

Reviewer #1: Yes

Reviewer #2: Yes

Reviewer #3: Yes

5. Review Comments to the Author

Reviewer #1: Review of Utilizing computer vision for facial behavior analysis in schizophrenia studies: A systematic review due to PLOS ONE:

The authors introduce a review of computer vision studies used to detect the schizophrenia disease in facial behavior.

Although the subject elaborated by authors is interesting, I have a number of observations on their work.

My observations are as follows:

(1) First of all, the abstract does not explain the challenges encountered in the task.

For example, it is not interesting to make that the reference used is only "google scholar "

and the words used is terms: 1.(“Computer Vision” or “Affective Computing”) AND “Schizophrenia” and 2. “Facial Expression” AND “Schizophrenia” AND "Computer Vision”.

(2) The author also say that they used the most relevant and up to-date publications. I did not understand the relevance criterion used (impact factor of article, ...). It is also interesting to mention the years of articles publication which it is better to mention up to-date publications.

(3) In addition, I don't understand why the authors used the articles written by

the same group of authors, or identified articles that were cited by the articles in the

primary search.

(3) The article is very poorly presented and the choice of titles of sections is very poorly expressed:

You used the main title 'Materials and methods' even though you did not use any material The section 'Searching methods' contains information which is not important and this information is repeated in the introduction and the abstract.

(4) Several information is missing such as the description of the Schizophrenia disease, indeed I see that the authors must mention part of the article to explain that.

For example, how emotions can identify this disease?

what emotions are used (primary, secondary, etc.)?

What are the negative emotions you mentioned?

How can machine learning methods analyze these emotions in patients?

(5) In Table 1. Overview of participant interviews:

I suggest to present other information like the year of the article, the emotions used...

(6) In Table 2. Overview of data processing and statistical analyses, the table is not clear, I suggest to subdivided the table on sub tables and make just the information related to each section example in section Type of the raw data, i suggest to make table regrouped by type of data(video, image (2D, 3D) and make other information like the databases used, the number of samples in datasets, the number of data used for learning and test..... .

(7) In general, the paper needs a deep review, made by a native English speaker.

(8) Along the paper, I found several tables included in the document where the contents of the tables are not clear.

(9) In a survey papers, it is interesting to make a comparison between methods using the rate of classification values, precision...

(10) Finally, I see that is necessary to reorganize the paper by finding a way to regroup papers in sub-groups. Example, according to the machine learning methods, the emotions, the databases used, the classes...

Reviewer #2: The paper focuses on the goal of providing objective measures for the evaluation and diagnosis of schizophrenia.

In particular, it deals with utilizing computer vision and machine learning to measure facial movements.

It provides a systematic overview of computer vision for facial behaviour analysis in schizophrenia studies, its evolution, the clinical findings, and the corresponding data processing and machine learning methods.

As a general consideration, I don't like systematic reviews. I prefer survey manuscripts that provide an overview depending on the confidence of the authors with the subject and independent from the queries on google scholar.

Anyway, the following comments are independent from this initial consideration.

While reading the manuscript, especially in the first sections, it seems like the authors lost the focus of the paper stated in the title and in the abstract. I would have expected to start from an introduction describing how different aspect of clinical diagnosis have been faced by computer vision methods and I read a description of how papers have been selected and a dissemination about how interviews have been carried out. I found of interest from row 128.

In table 2 a column describing the goal of each paper should be added. Some papers stopped at lower-level analysis and leave to human the diagnosis. Others one tried to provide a higher-level outcome (pathological /not pathological). This is an interesting aspect in my opinion that should emerge.

As a general comment, Authors should consider their manuscript as a guideline for researchers facing this topic for the first time and they should provide any useful information to get started in using computer vision for schizophrenia diagnosis.

A graphical representation of the most interesting approaches can be help to understand the cutting-edge works.

On the other hand, I found very interesting the discussion provided by authors.

Minor comments:

Figure 1 has low quality.

References to following papers should be added.

[1] Leo, M., Carcagnì, P., Mazzeo, P. L., Spagnolo, P., Cazzato, D., & Distante, C. (2020). Analysis of facial information for healthcare applications: A survey on computer vision-based approaches. Information, 11(3), 128.

[2] Thevenot, Jérôme, Miguel Bordallo López, and Abdenour Hadid. "A survey on computer vision for assistive medical diagnosis from faces." IEEE journal of biomedical and health informatics 22, no. 5 (2017): 1497-1511.

Reviewer #3: Although the article is interesting and could be deemed useful for the research community, I do have it presents two major problems that should be addressed:

1) For a systematic review, the keywords used in the search are of utmost importance. I believe that there is no clear explanation of why these words were selected, and others such as "face recognition", "face analysis" or "face emotion" are left out. At least an explanation of how the keywords were selected and a possible exploratory search around the terms would be needed.

2) the conclusions and discussion are technically shallow. Although the usefulness of the studies is discussed sufficiently, there is no clear learning about what technical approaches would be useful for (semi)automatic assessment of schizophrenia, and the choice and analysis is left for the readers. Also there is no discussion on the latest advances of computer vision and its possible application to schizophrenia predition, or the real challenges and opportunities on the field, which mostly refer to data availability and privacy and data security constrains.

6. PLOS authors have the option to publish the peer review history of their article (what does this mean?). If published, this will include your full peer review and any attached files.

Reviewer #1: **Yes: **Siwar yahia

Reviewer #2: No

Reviewer #3: No

---

## [Author Response · Author response to Decision Letter 0]

17 Mar 2022

Response to reviewers:

Reviewer #1:

Review of Utilizing computer vision for facial behavior analysis in schizophrenia studies: A systematic review due to PLOS ONE:

The authors introduce a review of computer vision studies used to detect the schizophrenia disease in facial behavior. Although the subject elaborated by authors is interesting, I have a number of observations on their work.My observations are as follows:

(1) First of all, the abstract does not explain the challenges encountered in the task. For example, it is not interesting to make that the reference used is only "google scholar ". and the words used is terms: 1.(“Computer Vision” or “Affective Computing”) AND “Schizophrenia” and 2. “Facial Expression” AND “Schizophrenia” AND "Computer Vision”.

Reply 1:> Thanks for raising this issue. We have updated our search terms to cover a broader definition of facial behaviors and added clarity by reducing two searches to one with the new search term:

(“Facial emotion” OR “Facial Expression” OR "facial analysis" OR "Facial Behavior" OR "facial action units") AND “Schizophrenia” AND "Computer Vision”

We also added PubMed as a second search engine, where no results were found. We have updated the PRISMA flow diagram on page 4, the corresponding sentences in Methods section of Abstract and in ‘Searching methods” (page 4 line 72-78). We added:

“We conducted a literature search for publications published before (including) December 2021, on Google Scholar and PubMed in February of 2022 using the following search terms: (“facial emotion” OR “facial expression” OR "facial analysis" OR "facial behavior" OR "facial action units") AND “schizophrenia” AND "computer vision”. Multiple synonyms and sub-categories of facial behaviors were used in the first keyword set to cover a broad definition of facial behaviors, and the latter two key words were selected to limit our search with studies that used computer vision in schizophrenia.”

We have also segmented the abstract into four sections (Background, Methods, Results, Conclusion) to add clarity and to help explain more about the challenges. We summarized the results and identified the key challenges in conclusion with the following text:

“Seventeen studies published between 2007 to 2021 were included, with an increasing trend in the number of publications. Only 14 articles collected data during interviews, of which different combinations of evoked to passive, structured to unstructured interviews were used. Various types of hardware were adopted and different types of visual data were collected. Commercial, open-access, and in-house developed models were used to recognize facial behaviors, where frame-level and subject-level features were extracted. Statistical tests and evaluation metrics varied across studies. The number of subjects ranged from 2-120, with an average of 38. Overall, facial behaviors appear to have a role in estimating diagnosis of schizophrenia and psychotic symptoms. When studies were evaluated with a quality assessment checklist, most had a low reporting quality.

Despite the rapid development of computer vision techniques, there are relatively few studies that have applied this technology to schizophrenia research. There was considerable variation in the clinical paradigm and analytic techniques used. Further research is needed to identify and develop standardized practices, which will help to promote further advances in the field.”

(2) The author also say that they used the most relevant and up to-date publications. I did not understand the relevance criterion used (impact factor of article, ...). It is also interesting to mention the years of articles publication which it is better to mention up to-date publications.

Reply 2:> We apologize for the lack of clarity here. We included all relevant work found in the searches in the review. We have updated the sentence in abstract as:

“Relevant publications published before (including) December 2021 were identified and discussed”

We have described the years of articles publication in the section “Results” at page 5, line 115-118:

“These articles were published from 2007 to 2021, with the number of publications increasing over time: three completed before 2009, four completed between 2010 to 2014, and eight were completed subsequent to 2015.”

(3) In addition, I don't understand why the authors used the articles written by the same group of authors, or identified articles that were cited by the articles in the primary search.

Reply 3:> Thanks for raising the question. Although a systematic review was performed, when reading the discovered papers, it became clear that there were other notable papers worthy of inclusion, either from references or from the same research groups. We felt it would enhance the review to add such articles, rather than exclude them because a general search process of a systematic review did not discover the research on a first pass. We have modified our method accordingly in “Searching methods” at page 4, line 79-84:

“Based on the articles we found in the search, we conducted a secondary search to include other notable and relevant papers worthy of inclusion, which were written by the same group of authors, and included relevant articles which were cited by articles found in the primary search. The secondary process was adopted to enhance the review by including those relevant articles that were not discovered in the first search using a general search process.”

(3) The article is very poorly presented and the choice of titles of sections is very poorly expressed:

You used the main title 'Materials and methods' even though you did not use any material The section 'Searching methods' contains information which is not important and this information is repeated in the introduction and the abstract.

Reply 4: > Thanks for this feedback. We modified the title of several sections and subsections to add additional clarify for the reader, for example we changed the section 'Materials and methods' to ‘Method’, changed ‘Current use’ in ‘Results’ to ‘Study objectives and participant characteristics’, changed ‘Data handling’ in ‘Results’ to ‘Data pipelines’, ‘Types of raw data’ in ‘Results’ to ‘Types of visual data’.

We have also removed the detailed description of searching terms in abstract. Instead, we divided the abstract on page 2 into four sections (Background, Methods, Results, Conclusion) and updated it with more information. Please kindly refer to reply1 for the details.

(4) Several information is missing such as the description of the Schizophrenia disease, indeed I see that the authors must mention part of the article to explain that. For example, how emotions can identify this disease? what emotions are used (primary, secondary, etc.)? What are the negative emotions you mentioned? How can machine learning methods analyze these emotions in patients?

Reply 5: > Thanks for the suggestion. We added additional details about schizophrenia, specifically, adding additional description about the diagnosis and the related challenges in the first paragraph in ‘Introduction’ on page 2 with the following text:

“Schizophrenia is a severe psychiatric disorder with a lifetime prevalence of approximately 0.48% [1]. This condition is slightly more common in males [2], appears generally during early adulthood [3], and causes significant social and functional impairment. In 2013, schizophrenia was thought to have an annual economic burden of $155 billion in the United States [4]. Since the identification of schizophrenia in the late 1800s, significant efforts have been made to characterize symptoms of the disorder.

…

Schizophrenia is an illness that demonstrates heterogeneity in its symptoms from person to person, and each of these symptom categories can vary vastly by presentation leading to overlap with other diagnoses. Additionally, schizophrenia has a heterogeneous longitudinal course, with some individuals having a relapsing remitting course, others chronic symptoms, and others with symptoms followed by remission [6].

Currently, the diagnosis of schizophrenia is based on the self-report of the patient, what the interviewer observes, and collateral information, all of which can be highly subjective. Reducing subjectivity in establishing the diagnosis of schizophrenia is necessary from both a research perspective (to ensure treatments work for people with the same underlying condition) and a clinical perspective. Clinically, many people with schizophrenia have a lack of awareness that they have an illness [7], and those with poor insight may be at risk for nonadherence to antipsychotic medications and other negative outcomes [8].”

Various emotions and facial actions units were utilized previously with manual identification. We have discussed some examples of variation of the expression in anger/sadness and reduction in happiness in the third paragraph in ‘Introduction’. The negative emotions in this paragraph refers to anger and sadness. We have discussed the derived deficits such as responsivity and expressivity as well. We have updated that paragraph accordingly on page 3 line 41-43.

In addition, we identified the basic emotions (happiness, sadness, anger, surprise, disgust, fear, contempt, and neutrality.) on page 15 line 210-225 utilized in the surveyed studies. We have added more descriptions on the specific facial action units in Section ‘Data pipeline’-’Behavior recognition methods’ on page 15 with the following text:

“Three studies analyzed basic emotions [32,35,40], including emotional categories such as happiness [32,35,40], sadness [32,35,40], anger [32,35,40], surprise [40], disgust [40], fear [32,35,40], and neutrality [35,40]. Eight studies [36,37,41,42,44–47] focused on AU analysis, a proxy measure for underlying facial muscle movement, based on the earlier mentioned FACS. AUs including AU0 (Neutral Face [41,42]), AU1 (Inner Brow Raiser [36,37,41,42,44–47]), AU2 (Outer Brow Raiser [36,37,41,42,44–47]), AU4 (Brow Lowerer [36,37,41,42,44]), AU5 (Upper Lid Raiser [36,37,41,42,44]), AU6 (Cheek Raiser [36,37,41,42,44]), AU7 (Lid Tightener [36,37,44]), AU9 (Nose Wrinkler [36,37,44]), AU10 (Upper Lip Raiser [36,37,44]), AU12 (Lip Corner Puller [36,37,41,42,44–47]), AU 14(Dimpler [44–47]), AU15 (Lip Corner Depressor [36,37,44–47]), AU17 (Chin Raiser [36,37,44–47]), AU18 (Lip Puckerer [36]), AU20 (Lip stretcher [36,37,44–47]), AU23 (Lip Tightener [36,37,41,42,44]), AU25 (Lips part [36,37,41,42,44–47]), AU26 (Jaw Drop [44–47]), AU 43 (Eyes Closed [41,42,45–47]), AU44 (Squint, [45–47]) AU45 (Blink [44]), and AU62 (Eyes Turns Right [45–47]). Other non-traditional AUs, such as smile, frown and sneer, were also mentioned in [45–47]. “

Various machine learning methods were utilized to identify facial behaviors. We have discussed the general approaches in the section ‘Introduction’ on page 3 line 47-54. those approaches used in the surveyed studies in section ‘Behavior recognition methods’ on page 15.

(5) In Table 1. Overview of participant interviews: I suggest to present other information like the year of the article, the emotions used...

Reply 6: > Thanks for the suggestion. We did not include them in (previous) Table 1 (now, Table 2) because we designed Table 1 (now, Table 2) to focus on the participant interview techniques. However, we also agree that the year of the articles and the emotions used in them are useful information to include. Hence, we have added the year information in (previous) Supplementary Table 1 ‘Detailed description and findings’ and changed that into the new Table 1: “Overview of the participants, objective types, descriptions and findings” on page 6-9. We have also included emotion information in Table 3 (previously Table 2) on page 12-14.

(6) In Table 2. Overview of data processing and statistical analyses, the table is not clear, I suggest to subdivided the table on sub tables and make just the information related to each section example in section Type of the raw data, i suggest to make table regrouped by type of data(video, image (2D, 3D) and make other information like the databases used, the number of samples in datasets, the number of data used for learning and test..... .

Reply 7: > Thanks for the suggestion and we apologize for the unclarity here. We have rearranged Table 3 (previously Table 2) on page 12-14 and regrouped into different segments based on the type of raw data.

Different subsections (like ‘types of raw data’, ‘behavior recognition methods’, etc.) in the ‘Data pipeline’ text section correspond to different sections or column(s) in Table 2.

We have updated the ‘Data pipeline’ section on page 11, line 174-177 to clarify the organization and the content of Table 3. The updated paragraph now reads:

“Table 3 summarizes how data was collected, processed, analyzed and reported in the 17 studies we surveyed. More specifically, the following aspects were included: (1) the types of data collected, (2) features calculated from the data, (3) and the corresponding statistical analyses and performance metrics used for reporting the results.“

We have also updated each subsection (highlighted in blue on page 15-17) so it’s more clear to see which column(s) in Table 3 were discussed in those subsections. In addition, we have added more description on statistical tests in “Evaluation methods’’ subsections:

“Statistical tests including t-test, analysis of variance were mainly adopted to evaluate the differences between different clinical groups. However, some studies did not report using any statistical tests on the features or the performances.”

We have moved the description of subjects into section “Study objectives and participant characteristics” and Table 1 on page 5-9 to give an overview of the studies.

(7) In general, the paper needs a deep review, made by a native English speaker.

Reply 8: > Thanks for the suggestions. We asked multiple native English speakers (the last three senior authors) to carefully review it. Please see the highlighted (in blue) texts in the pdf file for all the edits from the initial submission.

(8) Along the paper, I found several tables included in the document where the contents of the tables are not clear.

Reply 9: > We have updated Table 3 (previously Table 2) on page 12-14 and the corresponding descriptions and clarifications to add more information and clarity. Please kindly refer to the described description in Reply 7 (for comment 6).

We have also added the column ‘Year’ in Supplementary Table 1 (now Table 1 on page 6-9) and moved the column ‘Subject’ from Table 3 (previously Table 2) to Table 1. Please kindly refer to the described description in Reply 6 (for comment 5).

(9) In a survey papers, it is interesting to make a comparison between methods using the rate of classification values, precision...

Reply 10: > Thanks for the suggestion. We also agree that the comparison between different studies and methods would be interesting. However, a fair and meaningful quantitative comparison between different methods is difficult to make with the reported information found in the surveyed studies. Here are a few main reasons:

There is a lack of consensus on the selection of performance metrics. For example, for studies with classification tasks, two studies used AUC as metric, and the other two used accuracy. We have presented this point in more detail in the ‘Evaluation methods’ section on page 17.

No open access dataset could be used as a benchmark, and different datasets used in different studies vary in many aspects such as data type, subject size, demographic and diagnoses distribution. The huge differences between different datasets make it unfair to directly compare the methods across datasets. We have discussed this point in the second paragraph of the ‘Existing barriers and future directions’ section in Discussion on page 19, line 402-420.

We have instead compared the methods applied on the same dataset by the same research group in the third- and fourth-to-last paragraph in the ‘Existing barriers and future directions’ section on page 20, line 443-457. For example, using learned subject-level features might not necessarily be utterly superior to manually designed ones, which were supported by studies done by Bishay et al..

To clarify the above points, we have added the following text to the first paragraph in the ‘Existing barriers and future directions’ section on page 19, line 397-399 :

“In addition, different datasets used in different studies vary in many aspects such as data type, subject size, demographic, diagnoses distribution, and the selection of performance metrics.”

(10) Finally, I see that is necessary to reorganize the paper by finding a way to regroup papers in sub-groups. Example, according to the machine learning methods, the emotions, the databases used, the classes…

Reply 11: > Thanks for the suggestion. Due to the lack of consensus on the methods and databases, we felt it is difficult to properly group the papers into consistent groups across different aspects of the studies. Instead, we have grouped them based on the method they used in a specific aspect. For example, in Table 2, we have grouped the studies into “evoked”, “passive”, “both” and “unknown” based on the types of interview used in the studies. We have also grouped the studies into “descriptive”, “predictive” and “both” based on how they reported their results (please kindly find more discussion in the “Study objectives and participant characteristics” section on page 5). The main barrier that stops us grouping them consistently throughout the review manuscript is that studies fall into different groups when being looked at different aspects (ex. Studies that used “evoked” interviews might use either “descriptive” or “predictive” methods for reporting the results.)

We have updated the above discussions into the first paragraph in the ‘Existing barriers and future directions’ section in Discussion on page 19 with the following text:

“Consequently, it is difficult to compare the performances of the different methodologies evaluated, which adds an additional burden to the researchers who want to follow or replicate the previous studies.”

Reviewer #2:

C1: The paper focuses on the goal of providing objective measures for the evaluation and diagnosis of schizophrenia.In particular, it deals with utilizing computer vision and machine learning to measure facial movements. It provides a systematic overview of computer vision for facial behaviour analysis in schizophrenia studies, its evolution, the clinical findings, and the corresponding data processing and machine learning methods.

As a general consideration, I don't like systematic reviews. I prefer survey manuscripts that provide an overview depending on the confidence of the authors with the subject and independent from the queries on google scholar.

Anyway, the following comments are independent from this initial consideration.

Reply 1: > Thanks for raising the issue on systematic reviews vs. narrative reviews. We felt that we are trying to combine the advantages of both methods by providing an overview on how computer vision and facial behavior analyses are currently being used in schizophrenia studies, while leveraging the search engines like Google Scholar and PubMed to get a better coverage of the subject.

Please find our detailed replies below to all the other comments.

C2: While reading the manuscript, especially in the first sections, it seems like the authors lost the focus of the paper stated in the title and in the abstract. I would have expected to start from an introduction describing how different aspect of clinical diagnosis have been faced by computer vision methods and I read a description of how papers have been selected and a dissemination about how interviews have been carried out. I found of interest from row 128.

Reply 2: > Thanks for the suggestion. We have segmented the abstract into four sections (Background, Methods, Results, Conclusion) to add clarity and to help explain more about the challenges. We summarized the results and identified the key challenges in conclusion with the following text:

“Seventeen studies published between 2007 to 2021 were included, with an increasing trend in the number of publications over time. Only 14 articles used interviews to collect data, of which different combinations of passive to evoked, unstructured to structured interviews were used. Various types of hardware were adopted and different types of visual data were collected. Commercial, open-access, and in-house developed models were used to recognize facial behaviors, where frame-level and subject-level features were extracted. Statistical tests and evaluation metrics varied across studies. The number of subjects ranged from 2-120, with an average of 38. Overall, facial behaviors appear to have a role in estimating diagnosis of schizophrenia and psychotic symptoms. When studies were evaluated with a quality assessment checklist, most had a low reporting quality.

Despite the rapid development of computer vision techniques, there are relatively few studies that have applied this technology to schizophrenia research. There was considerable variation in the clinical paradigm and analytic techniques used. Further research is needed to identify and develop standardized practices, which will help to promote further advances in the field.”

We have also added more description of schizophrenia, its diagnosis and the related challenges in the first paragraph in ‘Introduction’ on page 2-3 with the following text:

“Schizophrenia is a severe psychiatric disorder with a lifetime prevalence of approximately 0.48% [1]. This condition is slightly more common in males [2], appears generally during early adulthood [3], and causes significant social and functional impairment. In 2013, schizophrenia was thought to have an annual economic burden of $155 billion in the United States [4]. Since the identification of schizophrenia in the late 1800s, significant efforts have been made to characterize symptoms of the disorder. …

Schizophrenia is an illness that demonstrates heterogeneity in its symptoms from person to person, and each of these symptom categories can vary vastly by presentation leading to overlap with other diagnoses. Additionally, schizophrenia has a heterogeneous longitudinal course, with some individuals having a relapsing remitting course, others chronic symptoms, and others with symptoms followed by remission [6].

Currently, the diagnosis of schizophrenia is based on the self-report of the patient, what the interviewer observes, and collateral information, all of which can be highly subjective. Reducing subjectivity in establishing the diagnosis of schizophrenia is necessary from both a research perspective (to ensure treatments work for people with the same underlying condition) and a clinical perspective. Clinically, many people with schizophrenia have a lack of awareness that they have an illness [7], and those with poor insight may be at risk for nonadherence to antipsychotic medications and other negative outcomes [8].”

We have also added the discussion on previous related surveys of using computer vision in clinical diagnosis in “Introduction” on page 3, line 60-63. Please kindly see Reply 7 for details.

C3:In table 2 a column describing the goal of each paper should be added. Some papers stopped at lower-level analysis and leave to human the diagnosis. Others one tried to provide a higher-level outcome (pathological /not pathological). This is an interesting aspect in my opinion that should emerge.

Reply 3: > Thanks for raising this point. We completely agree that the goal of each paper is important to discuss. We have moved (previous) Supplementary Table 1 as the new Table 1 (on page 6-9) so the readers could have a better overview of the studies, including their different goals.

In the “Type” column, we have grouped the papers into “descriptive”, “predictive” and “both” based on the final goal of the papers. The “descriptive” studies used descriptive statistics to report schizophrenia phenomenology and stopped there without attempting to directly estimate the outcome; In contrast, the “predictive” studies utilized predictors to classify presence or absence of schizophrenia, certain schizophrenia symptoms, or to predict clinical outcome measure scores. We have updated the section “Study objectives and participant characteristics” in ‘Results' on page 5 correspondingly:

“The study objectives were divided into three types: 1) descriptive, meaning those that described schizophrenia phenomenology, 2) predictive, meaning those that utilized predictors to classify presence or absence of schizophrenia or those which predicted certain clinical outcome scores based on facial expressions, or 3) those which included both descriptive and predictive outcomes. Of the 17 included studies, eight were descriptive, one was predictive only, and eight were descriptive and predictive. The study objective type, participant characteristics, description of the studies, and a summary of the findings can be found in Table 1.”

C4: As a general comment, Authors should consider their manuscript as a guideline for researchers facing this topic for the first time and they should provide any useful information to get started in using computer vision for schizophrenia diagnosis.

Reply 4: > Thanks for pointing this out. We have updated our manuscript significantly in Abstract and Introduction to clarify and add more information about schizophrenia, the current use of computer vision in it, and the key challenges. We hope in this way we could promote the readers who are not familiar with the subject to learn and enter this area. Please kindly see Reply 2 for the detailed modifications in abstract and introductions.

C5: A graphical representation of the most interesting approaches can be help to understand the cutting-edge works.

Reply 5: > Thanks for the suggestion. We have added a new figure 2: “Visualization of the data pipelines” on page 11 (please find the separate image file (fig2.tiff) because PLOS ONE does not allow in-place figures) with a brief illustration of the representative approaches used in the studies.

On the other hand, I found very interesting the discussion provided by authors.

Minor comments:

C6: Figure 1 has low quality.

Reply 6: > Good catch on that. It seems the figure in the final pdf was compressed, hence the lower quality compared to the separate figure file (fig1.tif). We will contacted the journal to see whether this will affect the quality in the published version if accepted.

C7: References to following papers should be added.

[1] Leo, M., Carcagnì, P., Mazzeo, P. L., Spagnolo, P., Cazzato, D., & Distante, C. (2020). Analysis of facial information for healthcare applications: A survey on computer vision-based approaches. Information, 11(3), 128.

[2] Thevenot, Jérôme, Miguel Bordallo López, and Abdenour Hadid. "A survey on computer vision for assistive medical diagnosis from faces." IEEE journal of biomedical and health informatics 22, no. 5 (2017): 1497-1511.

Reply 7 >: Thanks for the suggestions. We have added the discussion on previous related surveys in “Introduction” on page 3, line 60-63:

“Previous reviews have investigated the usage of computer-vision-based facial information in medical applications in general [29,30], but they focused more on the specific technical facial analyses adopted than the complete processing and analyzing pipeline, and few schizophrenia studies were discussed in detail.”

Reviewer #3:

Although the article is interesting and could be deemed useful for the research community, I do have it presents two major problems that should be addressed:

1) For a systematic review, the keywords used in the search are of utmost importance. I believe that there is no clear explanation of why these words were selected, and others such as "face recognition", "face analysis" or "face emotion" are left out. At least an explanation of how the keywords were selected and a possible exploratory search around the terms would be needed.

Reply 1: > Thanks for raising this point. To address the lack of clarity in the selection of the keywords, we have updated our search terms to cover a broader definition of facial behaviors and to add clarity by reducing two searches to one with the new search term:

(“Facial emotion” OR “Facial Expression” OR "facial analysis" OR "Facial Behavior" OR "facial action units") AND “Schizophrenia” AND "Computer Vision”

We also added PubMed as a second search engine, where no results were found. We have updated the PRISMA flow diagram on page 4 (please kindly see fig1.tiff), the corresponding sentences in Abstract (first sentence in Methods) and in ‘Searching methods” (page 4 line 71-84). We added:

“We conducted a literature search for publications published before (including) December 2021, on Google Scholar and PubMed in February of 2022 using the following search terms: (“facial emotion” OR “facial expression” OR”facial analysis” OR”facial behavior” OR”facial action units”) AND “schizophrenia” AND”computer vision”. Multiple synonyms and sub-categories of facial behaviors were used in the first keyword set to cover a broad definition of facial behaviors, and the latter two keywords were selected to limit our search with studies that used computer vision in schizophrenia.”

2) the conclusions and discussion are technically shallow. Although the usefulness of the studies is discussed sufficiently, there is no clear learning about what technical approaches would be useful for (semi)automatic assessment of schizophrenia, and the choice and analysis is left for the readers. Also there is no discussion on the latest advances of computer vision and its possible application to schizophrenia prediction, or the real challenges and opportunities on the field, which mostly refer to data availability and privacy and data security constrains.

Reply 2: > Thanks for raising this issue. We agree that leaving all the choices and analyzing options to readers, especially who are not familiar with the topic, could be challenging and drives the potential entries away. However, we wanted to point out that, although we would love to provide a complete guideline in this subject, there is not enough evidence to support which interview techniques or technical approaches are the best.

The key barrier here is the difficulty in making a fair and meaningful quantitative comparison between different methods with the reported information found in the surveyed studies. Here are a few main reasons:

There is a lack of consensus on the selection of performance metrics. For example, for studies with classification tasks, two studies used AUC as metric, and the other two used accuracy. We have presented this point in more detail in the ‘Evaluation methods’ section on page 17, line 291-300.

No open access dataset could be used as a benchmark, and different datasets used in different studies vary in many aspects such as data type, subject size, demographic and diagnoses distribution. The huge differences between different datasets make it unfair to directly compare the methods across datasets. We have discussed this point in the second paragraph of the ‘Existing barriers and future directions’ section in Discussion on page 19, line 402-420.

We have limited our comparison between the methods applied on the same dataset by the same research group in the third- and fourth-to-last paragraph in the ‘Existing barriers and future directions’ section on page 20, line 443-457. For example, using learned subject-level features might not necessarily be utterly superior to manually designed ones, which were supported by studies done by Bishay et al..

To clarify the above points, we have added the following text to the first paragraph in the ‘Existing barriers and future directions’ section on page 19, line 395-401 :

“Because of the sensitive and potentially identifiable nature of facial data for patients with schizophrenia, none of the datasets mentioned in this survey are publicly available. In addition, different datasets used in different studies vary in many aspects such as data type, subject size, demographic, diagnoses distribution, and the selection of performance metrics. Consequently, it is difficult to compare the performances of the different methodologies evaluated, which adds an additional burden to the researchers who want to follow or replicate the previous studies.“

To further address the dilemma between the lack of starting point or recommendation for the prospective readers and the lack of ample evidence from the literature, we added our suggestions in ‘Existing barriers and future directions’ on page 19-20 to point out the areas where we believe there is potential, and make it clear that further research is needed, with the following text:

“In addition, many state-of-the-art methods often provide publicly available implementations, such as JAA-Net (Joint facial action unit detection and face alignment via adaptive attention) [56]. …

Bishay et al. [41] compared the performance of the manually designed facial behavior features from [45,47] with the data-driven ones and showed the manual ones could be better in some cases.

As described above in the results section, temporal dynamics of the facial behaviors were not effectively used neither in behavior recognition modeling nor in the final symptom/treatment output classification or estimation. The former might be easier to start with since there are temporal facial expression datasets publicly available, such as Annotated Facial-Expression Databases (AFEW) [60].”

Regarding the lack of discussion on the latest advances of computer vision and its possible application to schizophrenia evaluation, we added a few latest progress in computer vision in the same section on page 20 and pointed out that they could potentially improve the facial behavior recognition accuracy in general population:

“Recent progress in computer vision could help bring superior performance in facial expression recognition. Replacing the current computer vision models used in affective computing with better backbone neural networks like ConvNext [61] and new video classification frameworks like video vision transformer [62] could be a potential direction.”

Lastly, we wanted to clarify that we believe how to apply computer vision based facial behavior recognition in schizophrenia study is still an ongoing discussion, where the community is still debating on both the clinical and technical sides of the approach, mainly due to the complex social, ethical, financial issues involved in this topic, when compared to general image/video classification/segmentation tasks. Although improving the accuracy of facial behavior recognition is a crucial part of the future directions, we believe the challenges and opportunities in this field also lie in addressing the issues such as reaching consensus in interview techniques, in data collection/security/privacy, and in method/result reporting, while reducing the bias and unfairness in both participants recruiting and diagnosing.

---

## [Decision Letter · Decision Letter 1]

29 Mar 2022

Utilizing computer vision for facial behavior analysis in schizophrenia studies: A systematic review

PONE-D-21-39628R1

Dear Dr. Jiang,

We’re pleased to inform you that your manuscript has been judged scientifically suitable for publication and will be formally accepted for publication once it meets all outstanding technical requirements.

Kind regards,

Felix Albu, Ph.D.

Academic Editor

PLOS ONE

Additional Editor Comments (optional):

The decision is Accept. The authors should take into consideration the comments about tables and subsections in the abstract.

Reviewers' comments:

Reviewer's Responses to Questions

**Comments to the Author**

1. If the authors have adequately addressed your comments raised in a previous round of review and you feel that this manuscript is now acceptable for publication, you may indicate that here to bypass the “Comments to the Author” section, enter your conflict of interest statement in the “Confidential to Editor” section, and submit your "Accept" recommendation.

Reviewer #1: All comments have been addressed

Reviewer #2: All comments have been addressed

Reviewer #3: All comments have been addressed

2. Is the manuscript technically sound, and do the data support the conclusions?

Reviewer #1: Yes

Reviewer #2: Yes

Reviewer #3: Yes

3. Has the statistical analysis been performed appropriately and rigorously? 

Reviewer #1: Yes

Reviewer #2: Yes

Reviewer #3: Yes

4. Have the authors made all data underlying the findings in their manuscript fully available?

Reviewer #1: Yes

Reviewer #2: Yes

Reviewer #3: Yes

5. Is the manuscript presented in an intelligible fashion and written in standard English?

Reviewer #1: Yes

Reviewer #2: Yes

Reviewer #3: Yes

6. Review Comments to the Author

Reviewer #1: The authors have addressed most of my comments. In my view the paper has been improved and can be published now.

Reviewer #2: The revised version of the paper is largely better than the initial one. The authors addressed all raised comments. They should pay attention to tables that are out of margins and I suggest them to not put subsections in the abstract but to leave the text as it is (I mean no titles of paragraphs).

Reviewer #3: The authors have tried to address most of my previous concerns. Although I still feel that the critical discussion of which methods are useful for the assessment of schyzophrenia and which ones are less so is still mostly missing, and the search terms are somehow limited, I understand that this is somehow a systematic review that needs to be relatively focused.

In such a young and unexplored research field, I would have preferred a narrative review style where the authors take a stand on the methods and speculate about the challenges that need to be overcome, trusting their expertise to select the relevant studies to be mentioned, but this is just my opinion.

I believe the manuscript has some merit and could be published in its present form, with maybe a small round of edits to streamline the content and improve the text flow.

7. PLOS authors have the option to publish the peer review history of their article (what does this mean?). If published, this will include your full peer review and any attached files.

Reviewer #1: **Yes: **siwar yahia

Reviewer #2: No

Reviewer #3: No

---

## [Editor Report · Acceptance letter]

31 Mar 2022

PONE-D-21-39628R1

Utilizing computer vision for facial behavior analysis in schizophrenia studies: A systematic review

Dear Dr. Jiang:

I'm pleased to inform you that your manuscript has been deemed suitable for publication in PLOS ONE. Congratulations! Your manuscript is now with our production department.

Kind regards,

on behalf of

Dr. Felix Albu

Academic Editor

PLOS ONE